# A Systematic Review of COVID-19 Infection in Kidney Transplant Recipients: A Universal Effort to Preserve Patients’ Lives and Allografts

**DOI:** 10.3390/jcm9092986

**Published:** 2020-09-16

**Authors:** Smaragdi Marinaki, Stathis Tsiakas, Maria Korogiannou, Konstantinos Grigorakos, Vassilios Papalois, Ioannis Boletis

**Affiliations:** 1Clinic of Nephrology and Renal Transplantation, National and Kapodistrian University of Athens Medical School, Laiko Hospital, 11527 Athens, Greece; smaragdimarinaki@yahoo.com (S.M.); mariok_25@yahoo.gr (M.K.); laikneph@laiko.gr (I.B.); 2Independent Researcher, 12 Protopappa Avenue, 16345 Athens, Greece; gk_pediatr@yahoo.gr; 3Renal and Transplant Directorate, Imperial College Healthcare NHS Trust, London W12 0HS, UK; vassilios.papalois@nhs.net; 4Department of Surgery and Cancer, Imperial College London, London SW7 2AZ, UK

**Keywords:** COVID-19 infection, kidney transplantation, outcomes, immunosuppression, treatment

## Abstract

The coronavirus disease 2019 (COVID-19) pandemic has posed a significant challenge to physicians and healthcare systems worldwide. Evidence about kidney transplant (KTx) recipients is still limited. A systematic literature review was performed. We included 63 articles published from 1 January until 7 July 2020, reporting on 420 adult KTx recipients with confirmed COVID-19. The mean age of patients was 55 ± 15 years. There was a male predominance (67%). The majority (74%) were deceased donor recipients, and 23% were recently transplanted (<1 year). Most patients (88%) had at least one comorbidity, 29% had two, and 18% three. Ninety-three percent of cases were hospitalized. Among them, 30% were admitted to the intensive care unit, 45% developed acute respiratory distress syndrome, and 44% had acute kidney injury with 23% needing renal replacement therapy. From the hospitalized patients a total of 22% died, 59% were discharged, and 19% were still in hospital at the time of publication. Immunosuppression was reduced in 27%, discontinued in 31%, and remained unchanged in 5%. Hydroxychloroquine was administered to 78% of patients, antibiotics to 73%, and antivirals to 30% while 25% received corticosteroid boluses, 28% received anti-interleukin agents, and 8% were given immunoglobulin. The main finding of our analysis was that the incidence of COVID-19 among kidney transplant patients is not particularly high, but when they do get infected, this is related to significant morbidity and mortality.

## 1. Introduction

As the first wave of the coronavirus disease 2019 (COVID-19) pandemic is continuing with different effects in different countries, our knowledge about disease features and outcomes of this novel coronavirus in the general population has grown substantially [1].

Kidney transplant (KTx) recipients have been recently classified by the Center for Disease Control and Prevention (CDC) as a high-risk group for severe COVID-19 [2]. Emerging evidence suggests 10-fold higher rates of early case fatality rate (CFR) in transplanted patients compared to that in the general population (GP) [3,4], due to the immunocompromised status resulting in impaired immunological response to pathogens [5] and to the almost universal presence of comorbidities [6,7].

The body of literature regarding COVID-19 infection in kidney transplantation is growing every day; however, it comprises mostly case reports, small case series, and small cohorts.

There is a broad variation in studies among different countries across the globe. The reported KTx recipients are heterogeneous in terms of race, ethnicity, time from transplantation, and baseline status at the time of COVID-19 infection.

In this systematic review, we analyzed the data of all studies reporting on adult KTx recipients with confirmed COVID-19. We focused on the following: (1) transplant characteristics and patient’s baseline status at the time of COVID-19 infection, (2) major outcomes of COVID-19 infection, and (3) therapeutic interventions including modifications of immunosuppression and investigational agents used for COVID-19.

## 2. Materials and Methods

### 2.1. Search Strategy

A systematic search of the literature published from 1 January to 7 July 2020, including PubMed, Web of Science, Scopus, and the Cochrane Library was conducted. The search terms incorporated were: “COVID 19” OR “SARS CoV2” OR “Coronavirus” OR “2019-nCoV” OR “SARS-CoV-2” OR “SARS-CoV” AND “renal” OR “kidney” AND “transplant” OR “transplantation” AND “recipient” OR “patient”, using Boolean operators, wildcards, and special characters as described in Appendix A.

Authors K.G. and S.M. independently reviewed the titles and abstracts for inclusion. This systematic review was conducted in accordance with the Preferred Reported Items for Systematic Reviews and Meta-Analyses (PRISMA 2009) [8], and the flow diagram is depicted in Figure 1. The PRISMA checklist is shown in Appendix A.

### 2.2. Study Selection

We included the following types of articles: case reports, case series, case–control studies, cohort studies, and correspondence articles. We did not include other systematic reviews, editorials, and conference abstracts. Since almost all studies were observational with small numbers of patients, we included all that were in accordance with our inclusion and exclusion criteria.

According to the P.I.C.O Model for Clinical Questions in Systematic Reviews, our intended patient population (P) comprised all adult (over 18 years old) solid organ transplant recipients who had undergone either kidney transplantation only or multiorgan—including as per definition kidney—transplantation with COVID-19 from 1 January until 7 July 2020. The intervention-exposure (I) was COVID-19 infection confirmed by nucleic acid amplification technique (NAT). We did not have a comparison (C) group. All major adverse outcomes (O) of COVID-19 infection, i.e., hospitalization, intensive care unit (ICU) admission, mechanical ventilation (MV), acute kidney injury (AKI), acute respiratory syndrome (ARDS), and death, were recorded as were recovery and discharge.

### 2.3. Data Extraction

We collected and analyzed the following parameters: last name of the first author, city or region, country, sample size, infection rate, hospitalization, duration of hospitalization (days), ARDS, AKI, ICU admission and duration of stay in ICU, type of MV (invasive and non-invasive), discharge, recovery, death, and case fatality rate (CFR).

Patients’ demographics and baseline characteristics included: age; gender; type of donor: living, deceased, donor after brain death (DBD), donor after cardiac death (DCD); multiple organ transplantation including, as per definition, kidney transplantation; recipient of first, second, or third kidney transplant; time since transplantation. Serum creatinine levels and, if available, estimated glomerular filtration rate (eGFR) were recorded at three time points: baseline at admission, peak during hospitalization, and at discharge. Comorbidities recorded were arterial hypertension (HTN); diabetes mellitus (DM); cardiovascular disease (CVD); malignancy (solid tumors and hematologic malignancies); obesity (OB); chronic obstructive pulmonary disease (COPD); and chronic viral infections including human immunodeficiency virus (HIV), hepatitis B virus (HBV), and hepatitis C virus (HCV).

Baseline immunosuppressive agents and regimens were also recorded, as were modifications in immunosuppression (IS) consisting of discontinuation, reduction, or switch from one agent to another. Induction therapy consisting of anti-interleukin-2 (anti-IL-2) agent basiliximab or antithymocyte globulin (ATG) were recorded in those transplanted for less than one year.

COVID-19-targeted therapies recorded included: antivirals, hydroxychloroquine (HCQ), antimicrobial agents, corticosteroid boluses, anti-interleukin-6 (anti-IL-6) monoclonal antibodies, interferon (IFN), immunoglobulin, colchicine, and the anti-chemokine-receptor-type 5 (CCR5) inhibitor leronlimab.

### 2.4. Statistical Analysis

We used the Microsoft Office Version 2019 platform to extract, collect, and analyze data. Individual participant data (IPD) were used for all patients reported separately. Aggregated data were also used when information about individual patients with kidney transplantation and confirmed COVID-19 infection was not available. Continuous variables were reported as mean ± SD and/or median. Categorical variables were reported as count and percentage.

## 3. Results

### 3.1. Overview of Studies

Our initial search retrieved a total of 328 articles. An additional 6 records were identified through manual screening. After the removal of duplicates, using Endnote Online as citation manager, the remaining 175 studies were screened by title and abstract. Subsequently, 90 articles were removed, based on relevance and inclusion and exclusion criteria. Full-length text was assessed for eligibility in 85 articles. Another 22 studies were excluded due to duplicate or mixed population, aggregated or missing data, and articles in a language other than English.

Finally, 63 articles reporting on 420 KTx recipients with confirmed COVID-19 infection were included. There were 28 case reports, 14 case series, 1 case–control study, 1 cohort study, and 19 correspondence articles from a total of 18 countries (Table 1; Table 2).

There were 142 patients from the U.S., 113 from Spain, 71 from Italy, 25 from China, 22 from Belgium, 13 from Iran, 8 from the U.K., 5 from Turkey, 5 from Portugal, 4 from Germany, 3 from Poland, 2 from Korea, and 1 each from the Netherlands, France, Canada, Brazil, and Thailand, respectively.

For identification of the risk of bias, authors S.M. and K.G. independently performed a quality assessment of 63 studies. In case of disagreement, the problem was solved by discussion. The quality of case series included was assessed using the Joanna Briggs Institute’s (JBI) critical appraisal checklist for case series, which consists of 9 quality items. Studies with up to 4 positive responses were considered to have low quality, while those with 5 to 9 positive responses had high quality. The study quality of case reports was assessed with the JBI checklist for case reports. The range of the JBI scale is between 0 and 8, with a score of 0–3 denoting low quality and 4–8 denoting high quality. Accordingly, all studies were included in the analysis. The risk of bias assessment is shown in Appendix A.

### 3.2. Demographic and Baseline Characteristics, Clinical Outcomes, and Treatment of Kidney Transplant Recipients with COVID-19 Infection

#### 3.2.1. Patients’ Demographics and Baseline Characteristics

The mean and median age of the 169 patients for whom IPD were available were 55 ± 15 years and 55 years (range 21–80) respectively. The reported median age from available aggregated data ranged from 57 to 60 years. There was a male predominance of 67%.

The majority of KTx recipients (74%) had received a deceased donor transplant (DBD in 95% of cases), while 26% had been transplanted from a living donor. In 156 cases, the donor source was not reported.

Seven out of 420 patients (2%) were multiple organ transplant recipients: 2 received liver and kidney, 2 received heart and kidney, and 3 received pancreas and kidney, respectively.

From the total cohort, only 2% patients had undergone subsequent kidney transplantations. Median time from KTx to COVID-19 infection was 6.5 years (range 0–33) while 23% of the patients were transplanted for less than one year.

The majority of patients (81%) suffered from HTN, while only 12% had no comorbidities. The second most frequent associated medical condition was DM (36%), followed by CVD (21%), OB (15%), COPD (5%), malignancy (4%), and chronic viral infection (2%).

Out of 162 patients with comorbidities, 41% had one, 29% had two, and 18% suffered from three comorbidities.

Baseline renal function was relatively well-preserved with a median serum creatinine of 1.47 mg/dL in 91 patients. Baseline eGFR was assessed in few studies (16 out of 63) with a mean of 40 ± 23 mL/min/1.73 m^2^. Median peak serum creatinine during hospitalization was 2.2 mg/dL (range 0.62–10.94) and returned to 1.4 mg/dL at discharge.

The patients’ clinical features and outcome as well as management strategies are depicted in Table 3.

#### 3.2.2. Major Clinical Outcomes after COVID-19 Infection

Infection rate in our study ranged from 0.27% to 1.67% and was calculated in those studies where the number of the total cohort of KTx recipients was available.

From the total cohort, 93% of patients were hospitalized. Median duration of hospitalization was 16 days (range 1–100). From 391 hospitalized patients, 118 (30%) were admitted to the ICU. In 32 out of 118 patients, the median duration of the ICU stay was 8.5 days (range 1–34). Non-invasive mechanical ventilation (NIV) was applied to 7% and invasive mechanical ventilation (IMV) to 23% of patients. ARDS was reported in (175/391) 45% of patients. A substantial proportion of patients, (150/345) 44% developed AKI, with need for renal replacement therapy (RRT) reported in 23%. Death was recorded in (93/420) 22% of patients. Most patients, (232/391) 59% were discharged; 29 patients remained hospitalized when the studies were published, 14 of whom were still in the ICU.

Case fatality rates in the case series including more than ten patients ranged from 6% up to 67%.

#### 3.2.3. Baseline Immunosuppression and Modifications during COVID-19 Infection

The most frequently applied immunosuppressive regimen at baseline consisted of a calcineurin inhibitor (CNI), an antimetabolite, and corticosteroids (CS) in 73% of patients.

In total, 64% (147/230) of patients were receiving tacrolimus (TAC), 10% (18/176) cyclosporine (CsA), 68% (217/319) mycophenolic acid (MPA), 14% (26/184) everolimus, 4% (9/211) azathioprine (AZA), and a minority (<2%) of patients other agents such as belatacept, mizoribine, or leflunomide.

Overall, IS was reduced in 27% (97/357), discontinued in 31% (66/212), and remained unchanged in 5% (14/275) of patients. The most frequently discontinued drug was the antimetabolite in 91% (227/250) of patients. Calcineurin inhibitors were reduced in 32% (65/204) and discontinued in 58% (118/204) of patients. Switch from TAC or mammalian target of rapamycin inhibitor (mTORi) to CsA occurred in 7% (24/358) of patients. The mTORi was reduced in 7% (2/27) and discontinued in 67% (18/27) of cases.

#### 3.2.4. COVID-19-Targeted Therapies

The main agents used for COVID-19 infection were antivirals, antibiotics, hydroxychloroquine (HCQ), anti-IL monoclonal antibodies, and steroid boluses.

In total, 30% (123/414) of patients received antivirals. The most frequently used antiviral was lopinavir/ritonavir, administered to 76% (94/123) of those. Other antivirals included darunavir/ritonavir (13%), ritonavir-darunavir/lopinavir (4%), oseltamivir or arbidol (11%), umifenovir (7%), and darunavir/cobicistat (2%).

Hydroxychloroquine was administered to 78% (320/409) of patients.

The majority of patients (73%, 290/399) received antibiotics: azithromycin was administered to 53% (155/290) and other broad-spectrum antibiotics to 17% (50/290) of cases.

Corticosteroid (CS) boluses or dexamethasone were used in 25% (83/331) of patients.

Anti-IL agents were introduced in 28% (59/213) of patients with more severe illness; tocilizumab was the preferred agent in 56 of 59 patients.

Less frequently used agents were immunoglobulin in 8%, colchicine in 0.5%, interferon in 0.5%, and leronlimab in 1.4% of patients.

### 3.3. Subgroup Analysis

We separately investigated three patient groups: recent transplanted recipients, elderly patients, and those who died. Outcomes are depicted in Table 4.

## 4. Discussion

In this review, we focused on the baseline status of the patients at the time of acquiring the infection, on the major clinical outcomes, as well as on the therapeutic interventions.

### 4.1. Infection Rates

Emerging evidence suggests that kidney transplant recipients are not at particularly high risk of acquiring the infection. Infection rates in our review range from 0.27% to 1.67%, with the highest rate of 5% reported in a study from Spain [56], in a cohort of elderly (>65 years) recipients. Of note, the infection rate among younger recipients in the same cohort was at 0.8%. However, infection rates depend greatly on the number of tested individuals; therefore, it is impossible to draw definite conclusions.

### 4.2. Clinical Presentation

Most agree that presenting symptoms are similar to those of non-transplanted patients with fever (85%), dry cough (70%), myalgia (60%), and dyspnea (57%) being the most frequently reported symptoms [6,69].

In a substantial number of transplanted patients, mild and/or atypical initial presentation with less fever and dyspnea and predominantly gastrointestinal symptoms has been reported [3,20,64,68,70], suggesting need for increased vigilance.

### 4.3. Disease Course

Illness severity at presentation among KTx recipients may vary significantly, similar to the case in the general population. However, acute respiratory decompensation and rapid clinical deterioration have been described in hospitalized as well as outpatient KTx recipients at an average of 7–10 days after disease onset. [3,7,53,61,71]. Though the management of KTx recipients with mild symptoms as outpatients may be a reasonable option, given the lack of prognostic indicators for eventual deterioration and current evidence about acute decompensation, rapid testing and early hospitalization is advisable.

### 4.4. Disease Severity

The initial suggestion that an immunocompromised status would hypothetically limit a striking cytokine release and lead to a milder disease course [72] has been confuted by current evidence. On the contrary, kidney Tx recipients acquiring COVID-19 infection are at high risk of developing severe disease due in fact to their immunocompromised status.

The presence of at least one comorbidity is an almost universal finding in transplanted patients. In non-transplanted individuals, comorbidities have been associated with adverse COVID-19 outcomes [73].

### 4.5. Baseline Status of Kidney Transplant Recipients

The mean age of the KTx recipients was 55 ± 15 years, and they were predominantly male.

Regarding transplantation parameters, most patients were deceased donor recipients. Time from transplantation to COVID-19 infection varied greatly: median 6.5, range 0–33 years. Only 23% were recently transplanted.

Renal function at baseline was relatively well-preserved: mean serum creatinine was 1.68 ± 0.77 mg/dL and mean eGFR 39.7 ± 23 mL/min/1.73 m^2^.

### 4.6. Comorbidities

Our analysis confirmed the almost universal presence of at least one comorbidity in the KTx population: HTN was the most prevalent in 81% of patients, followed by DM in 36%, CVD in 21%, obesity in 15%, and COPD in 5%. Remarkably, a substantial proportion (29%) of patients suffered from two and another 18% from three comorbidities. Hypertension, diabetes, CVD, and COPD have been identified early during the COVID-19 outbreak as risk factors for adverse outcomes [74]. African–American race as well as obesity have been recently associated with more severe COVID-19 [74,75].

### 4.7. Major Outcomes

All major adverse clinical outcomes related to COVID-19 including death were more prevalent in KTx recipients [76]. We found a hospitalization rate of 93%. This may include a selection bias, since indeed most studies reported those patients who had the most severe disease course and were hospitalized. We also found prolonged hospital stay (median 16 days) and a high ICU admission rate of 30% among them.

A total of 30% of KTx recipients needed mechanical ventilation: 23% invasive (IMV) and another 7% required non-invasive mechanical ventilation (NIV) support, while 45% of KTx developed ARDS.

We found a high rate of AKI in 44% of KTx recipients, compared to 29% in critically ill COVID-19-infected patients of the GP and to 15% in COVID-infected patients in general [77]. A substantial proportion of those developing AKI in our study (23%) needed RRT. AKI was reversible in most cases who recovered. Regarding the etiology of AKI, direct damage of the proximal tubular epithelial cells by the severe acute respiratory syndrome coronavirus 2 (SARS-CoV-2) has been reported early during the outbreak [78] but could not be confirmed later on. AKI as a result of renal damage due to uncontrolled cytokine storm seems to be currently the most prevalent theory [79]. Furthermore, in a kidney transplant recipient, IS reduction or withdrawal may lead to acute allograft rejection. No renal biopsy was reported in any of the 420 analyzed recipients. In the absence of biopsy confirmation and given the fact that AKI resolved in most of them who recovered, the most plausible explanation is the occurrence of AKI in the setting of multiorgan failure in patients with a sole functioning kidney and preexisting chronic kidney disease (CKD).

The overall death rate in our analysis was 22%.

In 13 case series including more than 10 patients, CFR ranged from 6% up to 67%. The highest CFR was reported by Nair (U.S., 33%), Crespo (Spain, 50%), Abrishami (Iran, 67%), Akalin (U.S., 28%), and Bossini (Italy, 28%) [3,6,50,53,56].

Nair [6] and Akalin [3] reported on 10 and 36 KTx recipients from NY city with CFRs of 33% and 28%, respectively. The small numbers of patients, the racial diversity, and the fact that the reports come from the epicenter of the pandemic in the U.S., are all factors that may have contributed to the high CFR in this series.

The study by Crespo et al. [56] reports a CFR of 50% in a selected cohort of 16 transplanted patients with confirmed COVID-19 infection who were all above 65 years. Older age is a known risk factor for adverse outcomes in patients with COVID-19. Further risk factors in this cohort included frailty, obesity, and underlying heart disease.

Abrishami et al. [50] from Iran, reports a CFR of 67% in a cohort of predominantly young patients with well-preserved renal function and few comorbidities. There is no obvious cause for this inexplicably high CFR in this study.

In a recent study, Bossini and Alberici [53] analyzed 53 KTx recipients from Italy, 45 of whom were hospitalized. The CFR was 33% in hospitalized patients and 28% in the entire cohort also including outpatients.

At this point, it is important to underline the high rates of adverse outcomes associated with COVID-19 infection in KTx recipients. Compared to the outcomes of influenza in solid organ transplant (SOT) recipients, as described in a review by Mombelli et al. [80], all adverse outcomes were higher in those with COVID-19 infection: 93% vs. 70% hospitalization rate and 30% vs. 16% ICU admission, respectively. The most striking difference was in early CFR: 22% vs. 3–8%.

### 4.8. Modifications in Immunosuppression

Managing IS in a KTx recipient in the context of severe infection is a complex approach. Since immunological response to infections is reduced in immunocompromised hosts, there is rationale to reduce and even to temporarily withdraw IS in case of severe COVID-19 disease. One has to balance the benefit of at least partially “restoring” the immune response in order to save the patient’s life against the risk of losing the graft due to acute rejection.

In our analysis, there was a high rate of IS reduction or withdrawal. Total IS was reduced in 27% and completely withdrawn, with the exception of steroids, in another 31% of patients.

The antimetabolites should be discontinued first because of their effect on inhibiting T-cell function and proliferation [81]. We found a rate of antimetabolite discontinuation of 91%.

Though mTOR-inhibitors have antiviral potential [82], they have been associated with various types of lung injury [83] and should preferentially also be withdrawn. In our study, the rate of mTORi reduction was 7% and the rate of withdrawal 67%.

Regarding CNIs, the most common approach is to minimize doses, which has proven efficacy in severe viral or opportunistic infections [5]. CNIs were withdrawn in 58% and reduced in 32% of cases in our study. Specifically, for cyclosporine, there are in vitro studies demonstrating that CsA suppresses viral replication through the inhibition of cyclophilin and this effect could be also demonstrated for the virus SARS-CoV-2. Based on this theoretical benefit and with the fear of completely withdrawing IS, another approach is to switch the tacrolimus or mTORi-based regimen to CsA, which occurred in 7% of cases in our study.

As for corticosteroids, since all practices including withdrawal, reintroduction, dose reduction, maintenance or increase, switch from oral to intravenous, or administration of boluses had been applied, we recorded only patients who had taken boluses as discussed in the next section.

### 4.9. COVID-Targeted Therapies

The main pharmacological interventions for the treatment of COVID-19 infection included antivirals in 30% of patients, broad-spectrum antibiotics in 73%, hydroxychloroquine in 78%, tocilizumab in 26%, steroid boluses in 25%, and less frequently other anti-IL agents (Clazakizumab, Anakinra), colchicine, immunoglobulin, interferon, and the anti-CCR5, leronlimab.

From the 30% of KTx patients who received antivirals, the majority, 76% received the combination of protease inhibitors lopinavir/ritonavir. They interact with CNIs by dramatically increasing their levels and prolonging half-lives; if used concomitantly with CNIs, drastic dose reduction and prolonged dosing intervals of the CNI are mandatory, otherwise patients will be exposed to prolonged overimmunosuppression, with detrimental effects in critically ill individuals [84]. The safest approach is to completely withdraw CNIs if they are co-administered. Moreover, they induce QT prolongation which, especially in combination with HCQ or azithromycin, is additive and may lead to severe arrhythmia [85].

Remdesivir is a nucleotide analogue initially developed to treat Ebola virus [86]. It has not been used until the outbreak of the COVID-19 pandemic but has shown efficacy in the GP [87].

Regarding hydroxychloroquine, early reports suggested a role in reducing the viral load [88]. Since it is cheap and easily available, it has been applied broadly. In total, 78% of KTx recipients received HCQ. Current data do not further support the use of lopinavir/ritonavir and HCQ in hospitalized patients with COVID-19 infection. On July 4, the World Health Organization (WHO) announced the discontinuation of the two treatment arms (HCQ/Lop-Riton) of the SOLIDARITY Trial based on results of the interim analysis that showed no effect in terms of reducing mortality [89].

Corticosteroid boluses have been used in 25% of KTx recipients. Since they increase viral replication, they are not desirable at the first phase of COVID-19 infection. In critically ill patients, they have immunomodulatory effects [90]. Steroid boluses are recommended for patients with ARDS; furthermore, the RECOVERY trial has shown benefit of high doses of dexamethasone in patients under mechanical ventilation [91]. The most reasonable approach is to maintain the lowest possible CS doses in the first phase of the infection and to administer boluses in those who develop severe illness.

Since severe COVID-19 infection has been associated with a cytokine storm [92], there is rationale for the use of anti-IL agents. Monoclonal antibodies that inhibit cytokines were used in 28% of KTx recipients in our study. The most commonly used was the IL-6 receptor antagonist tocilizumab in 26% of them. Less frequently applied agents included clazakizumab, anakinra, and leronlimab.

Unfortunately, due to the small numbers of patients in the studies investigated, the different time points during the disease course at which therapies were applied, and the different time points at which studies have been published, it is impossible to draw conclusions about the impact of therapeutic interventions on outcomes.

### 4.10. Subgroup Analysis

Three patient groups, i.e., recent transplanted recipients, elderly patients, and those who died, were analyzed separately.

A total of 23% with available IPD were recent KTx recipients. They did not differ from the total cohort in means of baseline characteristics, with the exception of better renal function. In terms of outcomes, though nearly half of them had received ATG, they had lower rates of AKI (38% vs. 44%), ARDS (31% vs. 45%), and death (15% vs. 22%) compared to those in the entire cohort. This finding further confirms preliminary data suggesting at least not worse outcomes in this subgroup of KTx recipients.

Patients older than 65 analyzed, comprised 34% of those for whom information was available. Only 4% had no comorbidity vs. 12% in the total cohort, while baseline creatinine was higher, at 1.8 mg/dL. A substantial proportion (17%) of them were on mTORi-based IS at baseline. The ICU admission rate was 19% vs. 30% in the total population, indicating either healthcare resource unavailability or decision not to intervene due to frailty and comorbidities. The death rate was higher at 32% vs. 22%.

Those who died (93 out of 420) were older: 27 out of 55 were over 65 years and predominantly male (61%), and they had worse baseline renal function. All adverse outcome rates were strikingly higher: ICU admission at 58%, 62% had need for invasive mechanical ventilation support, 81% developed ARDS, and 58% had AKI. In terms of therapeutic intervention, they had higher rates of intravenous CS (43%) but not of tocilizumab (20%) administration.

### 4.11. Limitations and Need for Future Research

The major limitation of our analysis was that the included articles were case reports and case series, which are subject to selection and publication bias. Thus, it is uncertain whether the results of our systematic review can be extrapolated to the general KTx-recipient population. There were a limited number of patients from all over the globe with a broad diversity in terms of race, ethnicity, and country of origin, as well as transplant and clinical characteristics. Besides patients’ heterogeneity, there is also variation in outcomes, disease course, and management of transplanted patients described, according to the time of publication of the study. Moreover, therapeutic interventions varied among countries depending on local policies, ethical issues, and healthcare resource availability conjointly with the “total COVID-disease burden” of the specific country. Thus, we could perform only descriptive statistics, and no conclusion could be drawn about the impact of therapeutic interventions on outcomes.

In view of the absence of a commercially available vaccine in the near future and given the fact that COVID-19 is our new reality, especially for vulnerable patient groups such as KTx recipients, large registry data and targeted studies assessing the impact of therapeutic strategies are urgently awaited.

In conclusion, the main finding in our analysis is the high rate of all major adverse outcomes of COVID-19 infection in hospitalized KTx recipients.

## Figures and Tables

**Figure 1 jcm-09-02986-f001:**
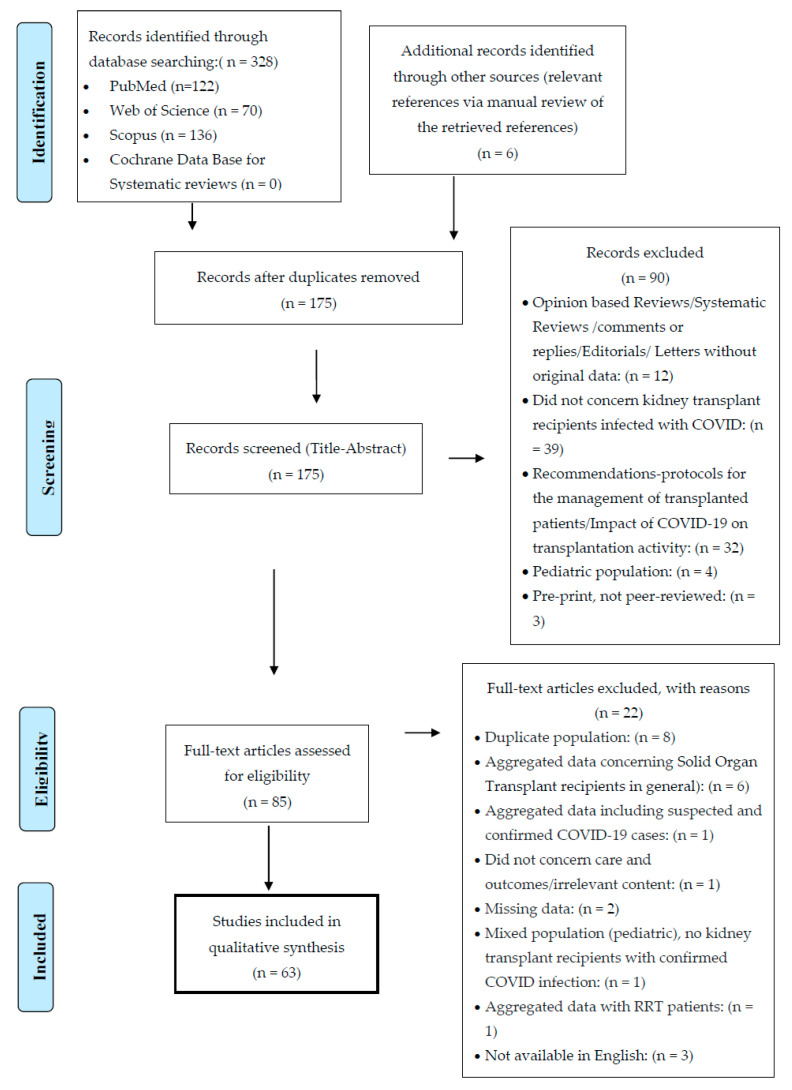
Flow diagram of the systematic literature search according to the Preferred Reported Items for Systematic Reviews and Meta-Analyses (PRISMA) statement. COVID-19: coronavirus disease 2019.

**Table 1 jcm-09-02986-t001:** Studies included in the analysis (Case Reports)

Case Reports				
Authors	Region/Country	Patients (No.)	Type of Study	Hospitalized
[9] Akdur et al.	Ankara, Turkey	1	Case Report	0
[10] Allam et al.	Fort Worth, TX, USA	1	Correspondence	1
[11] Bartiromo et al.	Florence, Italy	1	Correspondence	1
[12] Billah et al.	New York, NY, USA	1	Correspondence	1
[13] Bussalino et al.	Genoa, Italy	1	Case Report	1
[14] Chen et al.	Wuhan, China	1	Case Report	1
[15] Cheng et al.	Nanjing, China	2	Case Report	2
[16] Chenna et al.	Albany, NY, USA	1	Case Report	1
[17] Dirim et al.	Istanbul, Turkey	1	Case Report	1
[18] Fontana et al.	Modena, Italy	1	Case Report	1
[19] Gandolfini et al.	Parma, Italy	2	Correspondence	2
[20] Guillen et al.	Barcelona, Spain	1	Case Report	1
[21] Hasan Ahmad et al.	Ipswich, UK.	1	Correspondence	1
[22] Hsu et al.	Los Angeles, CA, USA	1	Case Report	1
[23] Huang et al.	Fuzhou, China	1	Case Report	1
[24] Jiang et al.	Wuhan, China	1	Correspondence	1
[25] Kates et al.	Seattle, WA, USA	1	Case Report	1
[26] Kemmner et al.	Munich, Germany	1	Correspondence	1
[27] Kim et al.	Daegu, Korea	2	Case Report	2
[28] Kocak et al.	Istanbul, Turkey	2	Case Report	2
[29] Kolonko et al.	Katowice, Poland	3	Case Report	3
[30] Kumar et al.	Chicago, IL, USA	1	Case Report	0
[31] Lauterio et al.	Milan, Italy	1	Correspondence	1
[32] Machado et al.	Sao Paulo, Brazil	1	Case Report	1
[33] Man et al.	Wuhan, China	1	Correspondence	1
[34] Marx et al.	Strasbourg, France	1	Correspondence	1
[35] Meziyerh et al.	Leiden, Netherlands	1	Case Report	1
[36] Namazee et al.	Semnan, Iran	1	Case Report	1
[37] Ning et al.	Hefei, China	1	Case Report	1
[38] Seminari et al.	Pavia, Italy	1	Case Report	1
[39] Serrano et al.	Hartford, CT, USA	1	Case Report	1
[40] Shingare et al.	Mumbai, India	2	Case Report	2
[41] SJ Antony et al.	El Paso, TX, USA	1	Case Report	1
[42] Suwanwongse et al.	New York, NY, USA	1	Case Report	1
[43] Tantisattamo et al.	Orange, CA, USA	1	Case Report	1
[44] Thammathiwat et al.	Bangkok, Thailand	1	Case Report	1
[45] Velioglu et al.	Istanbul, Turkey	1	Correspondence	0
[46] Wang et al.	Zhengzhou, China	1	Correspondence	1
[47] Xia et al.	Wuhan, China	1	Correspondence	1
[48] Xu et al.	Ottawa, Canada	1	Case Report	1
[49] Zhong et al.	Wuhan, China	1	Case Report	1

**Table 2 jcm-09-02986-t002:** Studies included in the analysis (Case Series)

Case Series						
Authors	Region/Country	Patients (No)	Type of Study	Hospitalized	Infection Rate (%)	CFR (%)
[50] Abrishami et al.	Tehran, Iran	12	Case Series	12	-	66.7
[3] Akalin et al.	Bronx, NY, USA	36	Correspondence	28	-	27.8
[7] Alberici et al.	Brescia, Italy	20	Case Series	20	1.67	25
[51] Banerjee et al.	London, UK.	7	Case Series	5	-	-
[52] Bosch et al.	Munich, Germany	3	Case Series	3	-	-
[53] Bossini et al.	Brescia, Italy	53	Case Series	45	-	28.3
[54] Chen et al.	Brooklyn, NY, USA	30	Case Series	30	-	20
[55] Columbia University	New York, NY, USA	15	Case Series	15	-	13.3
[56] Crespo et al.	Barcelona, Spain	16	Correspondence	15	4.93	50
[57] Devresse et al.	Brussels, Belgium	22	Case Series	18	1.83	9.1
[58] Fernandez-Ruiz et al.	Madrid, Spain	8	Correspondence	8	-	-
[59] Fung et al.	San Francisco, CA, USA	7	Case Series	5	-	-
[60] Maritati et al.	Ancona, Italy	5	Case Series	5	-	25
[61] Mehta et al.	New York, NY, USA	34	Correspondence	34	-	17.1
[62] Mella et al.	Turin, Italy	6	Case Series	6	-	-
[63] Montagud-Marrahi et al.	Barcelona, Spain	33	Correspondence	26	-	6
[6] Nair et al.	Hempstead, NY, USA	10	Case Series	9	-	33.3
[64] Rodriguez-Cubillo et al.	Madrid, Spain	29	Correspondence	29	-	20.7
[65] Silva et al.	Porto, Portugal	5	Case Series	4	-	-
[66] Trujillo et al.	Madrid, Spain	26	Case Series	26	1.04	23.1
[67] Zhang et al.	Wuhan, China	5	Case Series	5	-	-
[68] Zhu et al.	Wuhan, China	10	Case–control Study	10	0.33	10

CFR: case fatality rate

**Table 3 jcm-09-02986-t003:** Demographic and baseline characteristics, clinical outcomes, and treatment of kidney transplant recipients with coronavirus disease 2019 (COVID-19) infection.

Demographics	
Age (years, median) (*n* = 169)	55 (21–80)
Gender (male)	276/415 (67%)
Type of donor (deceased donor)	195/264 (74%)
Multiple organ transplant recipients	7/420 (2%)
Repeat KTx	7/420 (2%)
Time from KTx (years, median)	6.5 (0–33)
Time from KTx ≤ 1 year	48/209 (23%)
Comorbidities (*n* = 326)	
HTN	81%
DM	36%
CVD	21%
Obesity	15%
COPD	5%
Malignancy	4%
Chronic viral infection	2%
Renal function	
Baseline sCr (mg/dL, median) (*n* = 91)	1.47 (0.62–5.09)
Peak sCr during hospitalization (mg/dL, median) (*n* = 74)	2.17 (0.62–10.94)
sCr at discharge (median) (*n* = 58)	1.45 (0.29–6.45)
Baseline Immunosuppressive regimen	
MPA/AZA + CNI ± CS	136/186 (73%)
Hospital admission (*n* = 420)	93%
Duration of hospitalization (days, median) (*n* = 104)	16 (1–100)
Admission to ICU	118/391 (30%)
Duration of ICU stay (days, median) (*n* = 32)	8.5 (1–34)
Type of Ventilation	
NIV	27/379 (7%)
IMV	88/379 (23%)
ARDS	175/391 (45%)
AKI	150/345 (44%)
RRT	34/150 (23%)
Immunosuppression management	
IS discontinuation	66/212 (31%)
IS reduction	97/357 (27%)
Switch TAC or mTORi to CsA	24/358 (7%)
CNI tapering	65/204 (32%)
CNI withdrawal	118/204 (58%)
Antimetabolite withdrawal	227/250 (91%)
COVID-19 treatment	
Antivirals	123/414 (30%)
Lopinavir/Ritonavir	94/123 (76%)
HCQ	320/409 (78%)
Antibiotics	290/399 (73%)
Azithromycin	155/290 (53%)
CS (IV bolus or Dexamethasone)	83/331 (25%)
Anti-IL agents	59/213 (28%)
IV immunoglobulins	35/415 (8%)
Major outcomes	
Death	93/420 (22%)
Discharge	232/391 (59%)

Data are presented as the number/total number of available observations (percent) unless otherwise stated. Abbreviations: AKI: acute kidney injury; ARDS: acute respiratory distress syndrome; AZA: azathioprine; CNI: calcineurin inhibitor; COPD: chronic obstructive pulmonary disease; COVID-19: Coronavirus disease 2019; CS: corticosteroids; CsA: cyclosporine; CVD: cardiovascular disease; DM: diabetes mellitus; HCQ: hydroxychloroquine; HTN: hypertension; ICU: intensive care unit; IL: interleukin; IMV: invasive mechanical ventilation; IS: immunosuppression; IV: intravenous; KTx: kidney transplantation; MPA: mycophenolic acid; mTORi: mammalian target of rapamycin inhibitor; n: number; NIV: non-invasive ventilation; RRT: renal replacement therapy; sCr: serum creatinine; TAC: tacrolimus.

**Table 4 jcm-09-02986-t004:** Demographic and baseline characteristics, treatment, and clinical outcomes of specific subgroups of kidney transplant recipients with COVID-19 infection.

Characteristics of Patients with KTx for ≤1 year	
Age (years, mean ± SD) (*n* = 29)	53.9 ± 13.9
Gender (male)	18/34 (53%)
Baseline sCr (mg/dL, median) (*n* = 21)	1.24 (0.75–2.7)
Induction immunosuppression (*n* = 17)	
Antithymocyte globulin (ATG)	8/17 (47%)
Basiliximab	1/17 (6%)
Baseline Immunosuppression (*n* = 34)	
MPA + CNI ± CS	31/34 (91.2%)
IS discontinuation	9/34 (27%)
IS reduction	18/34 (53%)
Antimetabolite withdrawal	26/34 (77%)
COVID-19 Treatment	
Antivirals	7/34 (21%)
HCQ	18/32 (56%)
Azithromycin	3/32 (9%)
Corticosteroids (IV)	3/32 (9%)
Tocilizumab	3/32 (9%)
Hospital admission	32/34 (94%)
Admission to ICU	9/32 (28%)
ARDS	10/32 (31%)
AKI	12/32 (38%)
Death	5/34 (15%)
**Characteristics of patients ≥ 65 years**	
Age (years, mean ± SD) (*n* = 54)	71.6 ± 4.7
Gender (male)	45/70 (64%)
Baseline sCr (mg/dL, median) (*n* = 23)	1.8 (0.62–3.39)
Baseline Immunosuppression (*n* = 52)	
MPA/AZA + CNI ± CS	34/52 (65%)
mTORi-based regimen	9/52 (17%)
IS discontinuation	28/54 (52%)
IS reduction	17/54 (32%)
COVID-19 Treatment	
Antivirals	19/69 (28%)
HCQ	53/70 (76%)
Azithromycin	19/63 (30%)
Corticosteroids (IV)	18/70 (26%)
Tocilizumab	11/70 (16%)
Hospital admission	69/70 (99%)
Admission to ICU	13/69 (19%)
ARDS	32/69 (46%)
AKI	24/68 (35%)
Death	22/69 (32%)
**Characteristics of patients who died**	
Age (years, mean ± SD) (*n* = 37)	64.2 ± 9.1
Gender (male)	37/61 (61%)
Baseline sCr (mg/dL, mean ± SD) (*n* = 20)	2.14 ± 0.9
IS discontinuation	36/65 (55%)
IS reduction	18/51 (35%)
COVID-19 Treatment	
Antivirals	26/79 (33%)
HCQ	51/64 (80%)
Azithromycin	19/54 (35%)
Corticosteroids (IV)	32/74 (43%)
Tocilizumab	15/74 (20%)
Admission to ICU	47/81 (58%)
ARDS	62/77 (81%)
AKI	26/45 (58%)

Data are presented as the number/total number of available observations (percent) unless otherwise stated. Abbreviations: ATG: antithymocyte globulin.

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
