# Peer review of "A Systematic Review of COVID-19 Infection in Kidney Transplant Recipients: A Universal Effort to Preserve Patients’ Lives and Allografts"

_jcm, 2020, doi:10.3390/jcm9092986_

Round 1
Reviewer 1 Report
In this manuscript, the authors reported a systematic review of the literature on the clinical course and outcomes of COVID-19 in kidney transplant (KT) patients.
The authors should be commended for their efforts and for providing a detailed overview of the clinical consequences of COVID-19 in this fragile patient population.
The review is well written and is of interest for readers from both scientific and clinical perspectives.
The following points should be addressed by the authors to further increase the quality of the paper.
- Review methodology. Was a review protocol defined a priori (and published in the Prospero database) before conducting the review?
- Figure 1 (PRISMA flowchart): the information included in some boxes is not easily readable. Please edit the flowchart accordingly.
- Please better specify the PICOS framework of the systematic review in the methods section (i.e. study selection paragraph)
- Paragraph 2.5. may be removed, being a systematic review of the literature (no concerns regarding the ethics of the study)
- According to the PRISMA statement recommendations, a formal risk of bias assessment should be performed for all studies included in the qualitative analysis. Please provide the risk of bias assessment as a separate Table and add a paragraph in the results section.
- Please provide an overview of the diagnostic strategies used by the Authors of the included studies to diagnose COVID-19 infection in KT recipients. Ideally, the authors should report this information in a new Table.
- The authors included all the studies dealing with COVID-19 infected kidney transplant recipients. However, it would be also interesting to look at evidence published between Jan 2020 and July 2020 on the overall rate of COVID-19 infection in KT recipients. In other words, did the authors retrieve publications reporting the rate of infection at specific transplant Centres during the COVID-19 period? This information would provide the opportunity to assess the infection rate beyond the clinical consequences of the disease in COVID-19 positive patients.
In example, a study from Italy (Li Marzi V, Campi R, Pecoraro A, Serni S. Feasibility and Safety of Kidney Transplantation from Deceased Donors during the COVID-19 Pandemic: Insights from an Italian Academic Centre. Actas Urol Esp 2020. In Press) showed that KT from DBDs can be safely performed even during the COVID-19 pandemic, provided a logistical framework allowing to reduce the risk of nosocomial infection.
- In the discussion, the authors should acknowledge that a limitation of the current literature on COVID-19 infection in KT recipients relies on the lack of high-quality studies assessing the real incidence and prevalence of the infection among KT patients. Indeed, most studies are represented by case reports or small case series, with no “control” group, limiting the generalizability of the review findings to the “general population” of KT recipients.
Author Response
Response to Reviewer 1 Comments
Point 1: Review methodology. Was a review protocol defined a priori (and published in the Prospero database) before conducting the review?
Response 1: The systematic review was designed and conducted according to the PRISMA guidelines. However, a review protocol was not registered in the PROSPERO database due to a shortage of time.
Point 2: Figure 1 (PRISMA flowchart): the information included in some boxes is not easily readable. Please edit the flowchart accordingly.
Response 2: The information in the boxes of the PRISMA flowchart (Fig 1) has been revised accordingly.
Point 3: Please better specify the PICOS framework of the systematic review in the methods section (i.e. study selection paragraph)
Response 3: The PICOS framework was added to the Study Selection section.
Point 4: Paragraph 2.5. may be removed, being a systematic review of the literature (no concerns regarding the ethics of the study)
Response 4: Paragraph 2.5 has been removed from the revised manuscript.
Point 5: According to the PRISMA statement recommendations, a formal risk of bias assessment should be performed for all studies included in the qualitative analysis. Please provide the risk of bias assessment as a separate Table and add a paragraph in the results section.
Response 5: A paragraph describing the risk of bias was added in the Results section, Page 4 Line 167-175. Two Tables are provided as Supplementary Items 3 and 4.
Point 6: Please provide an overview of the diagnostic strategies used by the Authors of the included studies to diagnose COVID-19 infection in KT recipients. Ideally, the authors should report this information in a new Table.
Response 6: The typical diagnostic strategy followed by most authors included history of contact exposure; assessment of clinical symptoms, namely fever, dry cough or dyspnea; evaluation of chest X-ray or CT imaging findings; and positive nasopharyngeal swab for SARS-CoV-2 by nucleic acid amplification technique (NAT). We included only publications that fulfilled the criterion of COVID-19 infection confirmed by nucleic acid amplification technique (NAT), as described in Page 3 Line 111 in the Study Selection section, which is the most sensitive diagnostic method for COVID-19. We excluded studies with suspected COVID-19 cases.
Point 7: The authors included all the studies dealing with COVID-19 infected kidney transplant recipients. However, it would be also interesting to look at evidence published between Jan 2020 and July 2020 on the overall rate of COVID-19 infection in KT recipients. In other words, did the authors retrieve publications reporting the rate of infection at specific transplant Centres during the COVID-19 period? This information would provide the opportunity to assess the infection rate beyond the clinical consequences of the disease in COVID-19 positive patients.
In example, a study from Italy (Li Marzi V, Campi R, Pecoraro A, Serni S. Feasibility and Safety of Kidney Transplantation from Deceased Donors during the COVID-19 Pandemic: Insights from an Italian Academic Centre. Actas Urol Esp 2020. In Press) showed that KT from DBDs can be safely performed even during the COVID-19 pandemic, provided a logistical framework allowing to reduce the risk of nosocomial infection.
Response 7: As shown in Table 1 and mentioned in the discussion (Page 13 in the “Infection rates” section), infection rates are reported in 5 out of 22 case series included in the review, ranging between 0.27 to 1.67%, with the exception of the study from Spain conducted by Crespo et al., in which the infection rate reached almost 5% in KTx recipients older than 65 years old. Considering the fact that data on COVID-19 infection in KTx recipients is still scarce and that number of tested individuals varies between different countries, regions, institutions, it was beyond the purpose of this review to examine the safety of transplant activity during COVID-19 pandemic.
Point 8: In the discussion, the authors should acknowledge that a limitation of the current literature on COVID-19 infection in KT recipients relies on the lack of high-quality studies assessing the real incidence and prevalence of the infection among KT patients. Indeed, most studies are represented by case reports or small case series, with no “control” group, limiting the generalizability of the review findings to the “general population” of KT recipients.
Response 8: The limitations section was revised to: ‘The major limitation of our analysis was that the included articles were case reports and case series which are subject to selection and publication bias. Thus, it is uncertain if the results of our systematic review can be extrapolated to the general KTx recipient population. There was a limited number of patients from all over the globe with a broad diversity in terms of race, ethnicity, and country of origin, as well as transplant and clinical characteristics. Besides patients’ heterogeneity, there is also variation in outcomes, disease course and management of transplanted patients described, according to the time of publication of the study. Moreover, therapeutic interventions varied among countries depending on local policies, ethical issues, and healthcare resource availability conjointly with the “total COVID-disease burden” of the specific country. Thus, we could perform only descriptive statistics and no conclusion could be drawn about the impact of therapeutic interventions on outcomes.’
Reviewer 2 Report
Dear Authors: Your paper is timely and important. Your review of the literature is thorough. I very much appreciate how well written this is.
A few areas that warrant some claification:
Results:
- Page 7, line 178. Could you clarify what you mean by 2% of patients has undergone subsequent transplantation. Subsequent to what? Infection with COVID?
- Under comorbidities - OB? is that obesity?
Discussion:
1. Disease Course - The first sentence needs to be reworked. The content of the sentence is understood but the working is awkward.
2. Major Outcomes:
A. You state in the first sentence that all major adverse outcomes related to COVID10 were more prevalent in KTX recipients. Do you have a reference to support what the prevalence is of these major outcomes in non-transplant patients?
B. Page 14 Line 302. Please define GP - general population?
Author Response
Point 1: Page 7, line 178. Could you clarify what you mean by 2% of patients has undergone subsequent transplantation. Subsequent to what? Infection with COVID?
Response 1: By the term ‘subsequent transplantation’ we mean that the patient has undergone more than one kidney transplant in their lifetime.
Point 2: Under comorbidities - OB? is that obesity?
Response 2: By ‘OB’ we refer to obesity. We define the abbreviation earlier in the text, in Page 4 Line 129 in the ‘Data Extraction’ section.
Point 3: Disease Course - The first sentence needs to be reworked. The content of the sentence is understood but the working is awkward.
Response 3: The first sentence has been revised to: ‘Illness severity at presentation among KTx recipients may vary significantly, similar to the general population. However, acute respiratory decompensation and rapid clinical deterioration have been described in hospitalized as well as outpatient KTx recipients on an average of 7-10 days after disease onset.’
Point 4: You state in the first sentence that all major adverse outcomes related to COVID10 were more prevalent in KTX recipients. Do you have a reference to support what the prevalence is of these major outcomes in non-transplant patients?
Response 4: Major adverse events related to COVID-19 are more prevalent in kidney transplant recipients. For instance, according to the Johns Hopkins Coronavirus Resource Center, the overall case-fatality ratio even in regions with high incidence of COVID-19, including New York, Rio de Janeiro, UK, France, and Lombardi, is 7.52%, 7.08%, 12.42%, 8.85%, and 16.63%, respectively. Whereas the observed case-fatality ratio in kidney transplant recipients with COVID-19 infection according to our study is 22%.
Point 5: Page 14 Line 302. Please define GP - general population?
Response 5: By ‘GP’ we refer to general population. We define the abbreviation earlier in the text, in Page 1 Line 40 in the ‘Introduction’ section.
Reviewer 3 Report
Fig. 1 should be improved (truncated text in some frames).
Besides, it is a good paper. The topic is original and up-to-date. The main advantage of the manuscript is that the authors attempted to summarize available knowledge on that subject. The paper is well written and clear. Conclusions result from the data presented.
Author Response
Point 1: Fig. 1 should be improved (truncated text in some frames).
Response 1: The information in the boxes of the PRISMA flowchart (Fig 1) has been revised accordingly.